# Role of Oxygen Radicals in Alzheimer's Disease: Focus on Tau Protein

Anna Atlante [1,*], Daniela Valenti [1], Valentina Latina [2] and Giuseppina Amadoro [2,3,*]

1 Institute of Biomembranes, Bioenergetics and Molecular Biotechnologies (IBIOM)-CNR, Via G. Amendola 122/O, 70126 Bari, Italy; d.valenti@ibiom.cnr.it
2 European Brain Research Institute (EBRI), Viale Regina Elena 295, 00161 Rome, Italy; v.latina@ebri.it
3 Institute of Translational Pharmacology (IFT)-CNR, Via Fosso del Cavaliere 100, 00133 Rome, Italy
* Correspondence: a.atlante@ibiom.cnr.it (A.A.); g.amadoro@inmm.cnr.it (G.A.)

**Abstract:** Oxygen free radical burst is a prominent early event in the pathogenesis of Alzheimer's disease (AD). Posttranslational modifications of Tau protein, primarily hyper-phosphorylation and truncation, are indicated as critical mediators of AD pathology. This finding is confirmed by the high levels of oxidative stress markers and by the increased susceptibility to oxygen radicals found in cultured neurons and in brains from transgenic animal models expressing toxic Tau forms, in concomitance with a dramatic reduction in their viability/survival. Here, we collect the latest progress in research focused on the reciprocal and dynamic interplay between oxygen radicals and pathological Tau, discussing how these harmful species cooperate and/or synergize in the progression of AD. In this context, a better understanding of the role of oxidative stress in determining Tau pathology, and vice versa, primarily could be able to define novel biomarkers of early stages of human tauopathies, including AD, and then to develop therapeutic strategies aimed at attenuating, halting, or reversing disease progression.

**Keywords:** oxygen radicals; oxidative stress; Alzheimer's disease; Tau protein; mitochondria

## 1. Introduction: Why This Study?

Over a century after the first diagnosis, we cannot yet say with certainty what triggers Alzheimer's disease (AD), the infamous name of a scary disease that is difficult to understand.

A healthy human brain has tens of billions of neurons, specialized cells that process and transfer information through chemical and electrical signals. AD progressively destroys this communication mechanism, which leads to a loss of neuronal function and cell death. Regarding the pathogenesis of AD, the currently most accredited leading hypothesis involves two proteins, the so-called βeta-Amyloid (Aβ) and the Microtubule-Associated Protein (MAP) Tau [1,2]. Aβ peptide is formed from the breakdown of Aβ Precursor Protein (APP). One form of it, Aβ1-42, is highly toxic and pro-aggregant. In the brains of AD sufferers, abnormal levels of this peptide build up together to form neuritic plaques (NPs) that accumulate between neurons and destroy the normal cell function. The same goes for the Tau protein: it "detaches" from microtubules and assembles into "chains" that create clusters, or insoluble aggregates, inside neurons to form structures called neurofibrillary tangles (NFTs). NPs and NFTs in the brain are the characteristic histopathological features of the disease. Added to these hallmarks, there are overt oxidative stress (OS) and mitochondrial dysfunction [1–6].

In this framework, the studies by Amadoro et al. aimed at deciphering the functional role of the N-terminal domain of Tau, showed overexpression of some N-derived fragments located around different protease(s)-cleavage consensus sites of protein and found that high intracellular levels of Tau N-terminal fragments lacking the first 25 amino acids, such as the NH2-26-44 Tau fragment, evoke a powerful neurotoxic effect in primary hippocampal and cortical neurons [7,8]. This effect was supported by the extended stimulation of

extrasynaptic N-methyl-d-aspartate receptors (NMDAR), but the mechanism underlying the Tau fragment-induced excitotoxicity remains to be established.

Later on, Atlante et al. [9] proved that oxidative phosphorylation (OXPHOS) is dramatically impaired by this $NH_2Tau$ fragment, with the Adenine Nucleotide Translocator (ANT) as the unique mitochondrial target responsible for OXPHOS impairment. Considering that ATP generated into mitochondria through OXPHOS is primarily used in the cytosol and other intracellular organelles, it requires a fast mechanism for transporting ATP outside mitochondria and the ADP back inside. Relevantly, this is the main physiological role of ANT (for refs, see [10]). The impairment of the ANT explains the reduced availability of ATP in the cytosol—suggested as a cause of neuronal death mediated by the excessive and prolonged stimulation of NMDAR in AD (see [11]). In fact, it is believed that the abnormal activation of NMDAR causes the release of glutamate from the cell, due to electrogenic $Na^+$-coupled transporter reversals, and therefore excitotoxicity [12,13].

Several years after, we demonstrated that, in addition to $NH_2hTau$ fragment, which competitively inhibits ANT-1, Aβ 1–42 peptide also inhibits the ADP/ATP exchange in a non-competitive manner [14]. In addition, if these two toxic peptides are simultaneously added to the mitochondria, they collaborate by potentiating the ANT-1 dysfunction, which further aggravates the deficit of the mitochondrial ATP production at synaptic terminals [14].

Nevertheless, the mechanism modulating the interaction of the two AD fragments with ANT-1 still remained an object of speculation. By taking advantage of primary neuronal culture, namely cerebellar granule cells (CGCs), an in vitro system in which the main apoptotic steps were well characterized from a temporal and causative point of view (for refs, see [15,16]), we proposed a molecular mechanism of interaction between Aβ1-42, Tau fragment and mitochondria. In detail, starting from the assumption that: (i) a thiol(s) group(s) in ANT-1 molecules is/are target(s) of reactive oxygen species (ROS) (for refs, see [17,18]) and (ii) the ANT-1 is strongly inhibited by mersalyl (MERS), a reversible alkylating agent of thiol groups [19], in agreement with [20,21], we provided evidence that MERS—by blocking and protecting the -SH group—not only prevents the effect of $NH_2hTau$ (but not of Aβ1-42) on ANT-1 activity, but also revokes the toxic effect of $NH_2hTau$ on ANT-1 when Aβ1-42 is previously present [22]. Paradoxically, Aβ1-42, by interacting with ANT-1 in a non-competitive manner through ROS production, likely at the level of mitochondrial Complex I, protects the cell from the inhibitory effect of $NH_2hTau$ on ANT-1, as a consequence of thiol group/s oxidation and conformational changes in the carrier protein [22] (see Figure 1).

In this regard, we successively observed that ADP, added to cultured CGCs, also reduced the effectiveness of Tau in inhibiting ANT-1, as Aβ1-42 does, likely due to a protection of –SH group/s—presumably located at the ANT-1 active site (see Figure 1)—and involved in the interaction of $NH_2hTau$, but not of Aβ1–42, with ANT-1 [23].

It follows that the molecular mechanism underlying the pathological Aβ-$NH_2hTau$ interplay on ANT-1 in AD neurons involves: (i) thiol groups present at the active site (see [22]) and (ii) the ROS increase which oxidizes these—SH residues, as in [24], acting as modulators/blockers of $NH_2hTau$ fragment toxicity.

In this context, the convergence of Tau and Aβ on mitochondria in AD explains why the strategies adopted to modify the pathological forms of Aβ- or Tau- have not individually yielded promising results and, consistently, it highlights potential, new pathway(s) and target(s) for effective mutual therapeutic intervention of early dysfunction in AD. Interestingly, the crucial role of ANT-1 impairment in AD onset/progression suggests new approaches aimed to preserve/ameliorate mitochondrial function. This review aims to collect the latest progress in research focused on the interplay between ROS and Tau fragments and on their cooperative and/or synergic contribution to the progression of the disease.

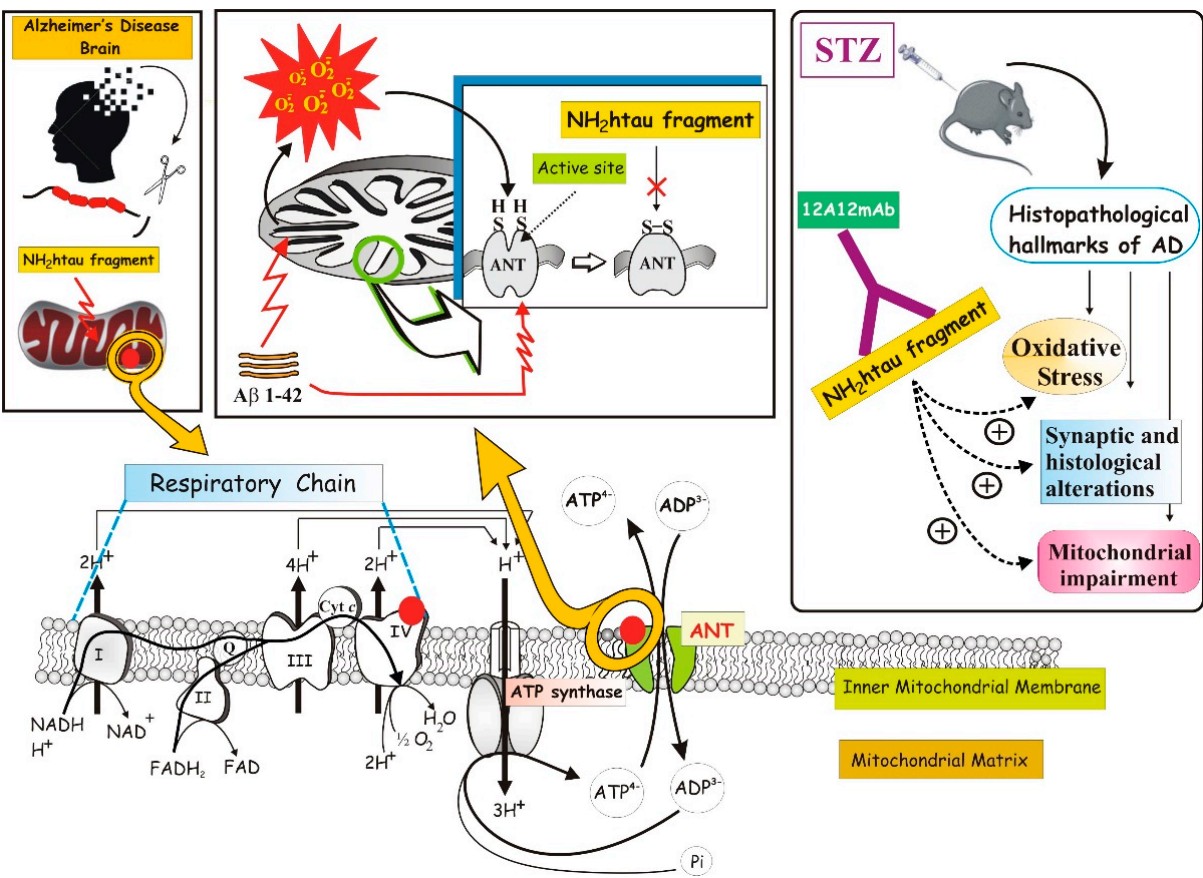

**Figure 1.** The effect of NH$_2$hTau fragment on mitochondrial Adenine Nucleotide Translocator (ANT-1) in Alzheimer's disease. The neuroprotective effect of Tau-fragment immunization, involving in part the modulation of oxidative stress (OS) and mitochondrial energetic deficits, is also depicted here.

## 2. Oxidative Stress, a Real Cruelty to the Brain

This chapter aims, not so much to describe the OS that—thankfully—is well known to all, but mainly to present, one by one, the performers, the extras and the main protagonists who will form the cast of this Review and which, in due course, will enter the scene.

It is a biological paradox: oxygen, the basis of life on this planet, is also reactive and toxic; therefore, when the oxidizing substances, including ROS, prevail or the antioxidant (AOX) substances are reduced, the cellular heritage undergoes continuous OS [25].

ROS, small and particularly aggressive particles, are formed incessantly in our organism as a waste product of natural metabolic processes that use oxygen as fuel to produce energy (oxidation). In practice, each free radical has lost the chemical "partner" to which it was linked due to metabolic processes or many other factors (pollution, UVA rays, prolonged stress, smoking, etc.) (see Figure 2). From that moment, it begins the search for a new "partner" from which to recover the missing electron. This search takes place in a spasmodic, indiscriminate, very fast way (we are talking about fractions of seconds!): in this desire for "coupling", the radicals are very warlike and attack fundamental parts of cells. To regain their stability, free radicals try to recover an electron from another molecule, which when "robbed" becomes incomplete and in turn will destabilize, transforming into a free radical, and then will "attack" another molecule and steal an electron, and so on. In this way, a chain of reactions is generated which spreads very quickly and can involve thousands of molecules, with harmful consequences for all the cells of the organism that age prematurely and that favor diseases and degeneration.

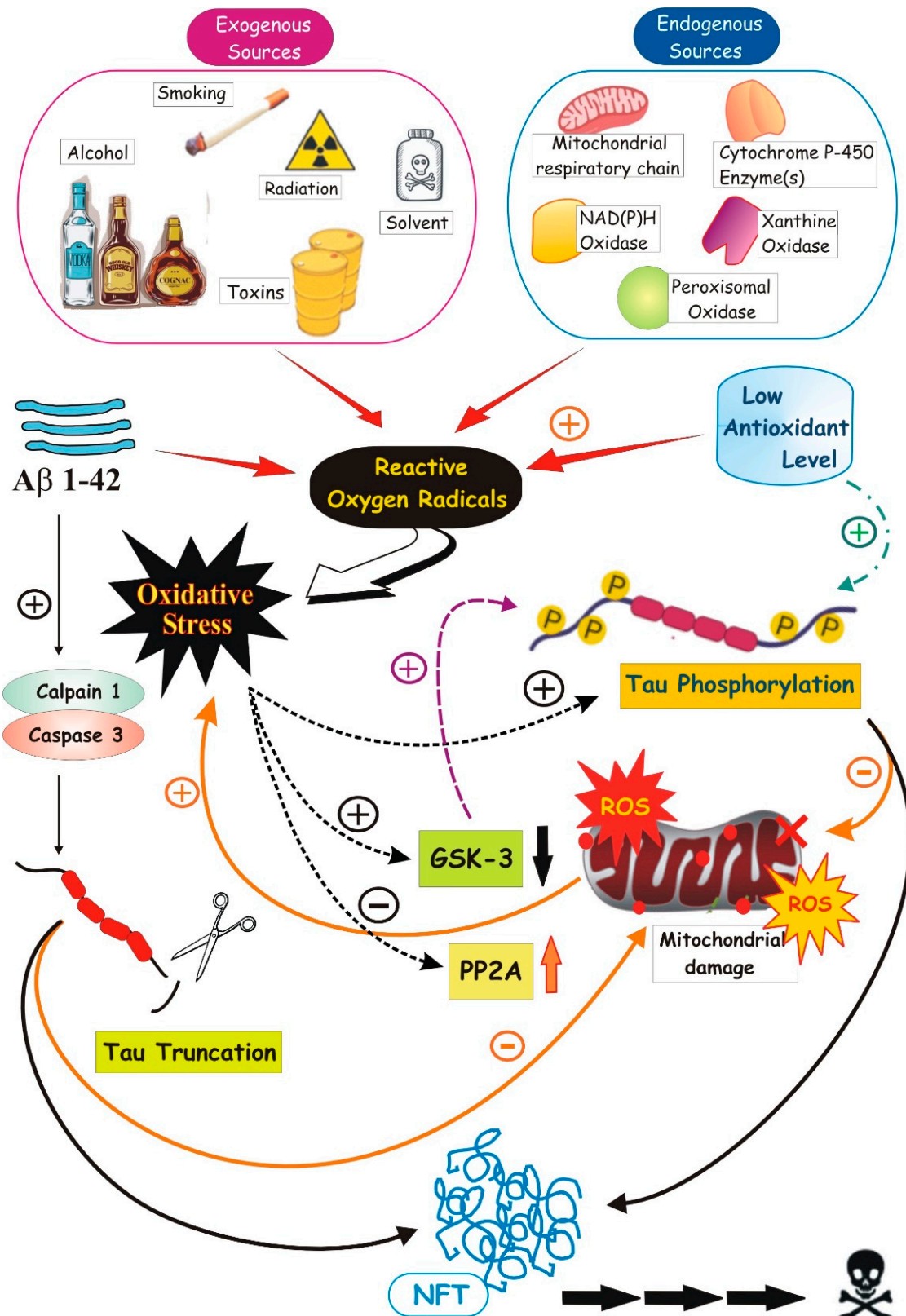

**Figure 2.** The picture presents data from several studies on pathological Tau forms, i.e., hyperphosphorylated and truncated Tau, with respect to OS and mitochondrial damage.

Fats and proteins, present on cell membranes but also on internal cell organelles and nucleic acids (DNA and RNA, genetic heritage), stored in the cell nucleus, represent the main targets of the oxidizing action of ROS (for refs, see [26]).

ROS are produced by our organism mainly at the level of the mitochondrial respiratory chain (mRC) (see Figure 1), consisting of the protein complexes I, II, III, IV and V. In the OXPHOS process, ATP is produced through the V complex, i.e., F1Fo-ATP synthase, using the proton gradient produced by complexes I, III and IV as electrons flow along the mRC. Regarding the ATP synthase, several authors reported a functional expression of the entire electron transfer chain, including the ATP synthase, in other cellular membranes (for refs, see [27]). These extra-mitochondrial sites synthesize aerobic ATP without the restraints imposed by the double membrane system of the mitochondrion. If we consider that the myelin sheath also generates aerobic ATP, this information adds a valuable piece of knowledge to the understanding of AD: since Aβ inhibits the formation of myelin [28], it follows that the loss of myelin in AD may be involved in a kind of vicious circle, by promoting further neuronal loss and disease progression.

About 1–2% of the electrons escape from the complexes and, interacting with oxygen, generate the superoxide anion ($O_2^-$) [29] (see Figures 1 and 2). $O_2^-$ is the primary ROS produced by metabolic processes. It interacts directly with other molecules through enzymatic or metal-catalyzed processes to generate secondary ROS (for refs, see [30]). The oxidation of proteins by ROS changes their properties and may increase their propensity to aggregate [31]. This property plays an important role in AD and other neurodegenerative diseases.

In addition to the mitochondrion, other sources of ROS are present in humans, including peroxisomal oxidase, cytochrome P-450 enzymes, NAD(P)H oxidase and xanthine oxidase (for refs, see [26] (see Figure 2)).

Regarding AOX, enzymatic or non-enzymatic [32], they are substances capable of counteracting, slowing down or neutralizing the formation of oxygen radicals that are formed following oxidation reactions [33]. The 'primary enzymes' are those enzymes that catalyze the transformations of ROS; the 'secondary enzymes', on the other hand, have the function of implementing antioxidant defenses [32], e.g., glucose-6-phosphate dehydrogenase regenerates NADPH, which a primary enzyme needs to function. Non-enzymatic AOX are divided into endogenous, i.e., produced by our organism, and exogenous, that is, those we gain from foods of vegetable origin.

Glutathione (GSH) is the most important intracellular defense against the deleterious effects of ROS. ROS oxidize GSH and the reduced form is regenerated by a NADPH-dependent reductase. The ratio of the oxidized form of glutathione (GSSG) to the reduced form, i.e., GSH, is a dynamic indicator of an organism's OS [34].

Particular attention should be paid to exogenous AOX since this is the most unpredictable component in the cellular redox balance. Vitamins C (ascorbic acid) and E possess high AOX effects. Ascorbic acid must be ingested from food (or supplements); tomatoes, pineapples, watermelons and all citrus fruits contain the highest amounts of vitamin C [32]. In contrast, vitamin E, a fat-soluble vitamin, is mainly present in vegetable oils, nuts, broccoli and fish. The use of AOX supplements containing multivitamins and minerals has grown in popularity among consumers, particularly in recent years. To name a few, carotenoids, a large class of tetraterpenes widely found in plants, also include ROS scavenging among their biological activities [35,36]. β-carotene, one of the most well-known carotenes, is mainly found in carrots, pumpkins, mangoes and apricots. Still, among the most studied polyphenols, curcumin has also received a lot of attention for nutraceutical applications, essentially for its powerful scavenging activity of $O_2^-$, $H_2O_2$, lipid peroxides and various RNS [37]. Curcumin also increases cellular GSH levels 3.

## 2.1. Bioenergetic Efficiency of the Brain Is Vital for Normal Functioning of the Central Nervous System (CNS)

OS, in addition to being responsible for aging and tissue degeneration, is the arch enemy of our health. It not only damages the skin, hair and nails that we take care of so much, but all our tissues, including nerve cells, the well-being of which we tend to neglect.

The brain—corresponding to 2% of the total body mass—is the organ of the body that ages faster and more significantly than all other tissues in the body [38]. The reason for this phenomenon is intrinsically linked to biochemistry and brain function. In fact, neurons, the main cells of which the brain is composed, are post-mitotic, that is, they do not duplicate or regenerate (they do so only in limited areas, through neurogenesis, which, however, has very little impact in terms of replacement). As a result, once they die, they are not replaced by new cells. Furthermore, it should be remembered that the brain is a structure with a high energy metabolism; therefore, in addition to producing large quantities of energy, essentially used to maintain the neuronal membrane potential (see [39]) and, in addition, for the synthesis/release of neurotransmitters and for axoplasmatic transport (see [39]), it uses large quantities of oxygen (1/3 of the oxygen we breathe is used by the brain) and, therefore, produces many free radicals capable of causing irreversible damage at the cellular level [40]. This is the reason why the brain presents itself as a fertile battlefield: the nerve cell is attacked and damaged on several fronts. The damage caused can range from a simple functional deficit to the death of the nerve cell, processes that must be related to disturbed oxygen metabolism and impaired mitochondrial activities [41].

Another aspect that should not be underestimated is that about 50% of the dry weight of the brain is made up of lipids [42]; they are easily oxidized and undergo structural and functional damage. Specifically, to promote very rapid nerve conduction, our nerves are also surrounded by a myelin sheath, which contains many lipids and is therefore also exposed to the attack of free radicals. The myelin sheath is characterized by 75–80% lipids, and membranes and neurons are also rich in polyunsaturated fatty acids [43]; therefore, they are also exposed to the aggression of ROS, which also attack proteins, essential components of membranes and neurotransmitters and also of the immune defenses.

Not content, ROS reach the nerve cell command center, the nucleus, which contains the delicate genetic equipment.

However, that is not all: nerve cells—i.e., the neurons—have a very high concentration of mitochondria that work at full capacity in order to ensure sufficient energy for the long nerve branches (axons and dendrites); mitochondria represent the main generators but also targets of ROS which damage their wall and reduce their bioenergetic efficiency, i.e., reduce the neuron's ability to produce ATP, or to have the energy that allows it to increase its self-healing capabilities.

Furthermore—and this is almost a paradox—the brain has a very low concentration of endogenous AOX (protein and non-protein). For example, the levels of GSH, superoxide dismutase (SOD) and catalase are about 1/5 of those in the liver. In fact, therefore, the brain is by its nature extremely exposed to OS and consequently ages more prematurely than other tissues.

AD is characterized by a huge increase in ROS and, thus, antioxidant therapeutic strategies are currently exploited in the treatment of this disorder, as demonstrated by the results of ongoing experimental and clinical research (for refs, see [44]).

However, we deal with this in the next chapter.

## 3. Characteristics of Alzheimer's Disease: An Overview

You find yourself disoriented on a street corner without knowing why you are there, you forget what the keys are for, or worse, you do not recognize the loved ones and the affections of a life: one in three people at the age of 90 have Alzheimer's or another senile dementia. We are facing a generational pandemic. One thing is clear about AD: lifestyle matters a lot. Exercise is important, just as a diet that is too high in cholesterol. However, what really matters is using your brain, if you do not use it, it atrophies. Cognitive gymnastics are good for those suffering from senile dementia, so much so that where there is a lot of illiteracy, AD is 3–4 times more common.

The pathology is complex due to its heterogeneity of both causes and symptoms (see [2]). Most cases of AD are sporadic due to unknown causes, not hereditary. Less than 10% of AD cases are caused by mutations in certain genes, such as the APP, Tau, Presenilin 1 (PS1), Presenilin 2 (PS2) and polymorphisms in Apolipoprotein E (ApoE). Exactly what causes AD is still a mystery to science: the evidence suggests a combination of environment, genes and lifestyle. AD is also a progressive disease, which means it gets worse over time. It starts with changes in the brain that are unnoticeable to the affected person. The obvious symptoms—first of all memory loss—only appear after years of brain changes and occur because the neurons of parts of the brain involved in thinking, learning and memory have been damaged. Along with the clinical manifestations, there are distinctive neuropathological signs in the AD brain, mainly in the hippocampus and cortex (for refs, see [45]). As the disease progresses, neurons in parts of the brain are also affected that allow a person to perform basic bodily functions, such as walking and swallowing. AD is ultimately fatal.

NPs and NFTs are the main histopathological lesions that define the AD brain [46]. The definitive diagnosis of AD can only be made by examining post mortem brain tissue based on the presence of extracellular NPs formed by Aβ peptides, intracellular NFTs consisting of hyperphosphorylated Tau (pTau) and truncated Tau protein, Aβ heaps in blood vessels, neuronal loss and synaptic and significant atrophy in specific brain regions involved in cognitive function (hippocampus, entorhinal and frontal cortex) [47].

At the moment, the development of effective therapies for the treatment of AD is of utmost urgency if we consider that (i) the disease is severe, (ii) the number of patients is continuously increasing and especially (iii) the therapies available for AD are only symptomatic (see [48]), i.e., they fail to ameliorate cognition. Consistently, the amyloid hypothesis—in vogue since 1984 [48–51]—has been challenged so much so that the number of anti-amyloid trials was reduced significantly in 2019, due to a series of clinical failures which question further development of Aβ-targeting drugs for AD treatment. The failing pharmacological aspect is aggravated by the fact that the symptoms due to the disease occur when the damage in the brain has already accumulated for 5 or 10 years. At that point, it is difficult to reverse the disease, leading to massive tissue loss throughout the brain. Added to this is the fact that the preclinical evaluation of AD drugs is hampered by the lack of validated experimental models and transgenic animal models which do not adequately represent the pathology and clinical progression of AD.

### 3.1. Misfolded Proteins in the Mechanism of Neurodegeneration

APP, a protein placed in the cytoplasmic membranes of nerve cells, is primarily expressed in the nervous system, where it has a very important role in the neurodevelopment in modulation of synaptic activity, closely linked with the phenomena of brain plasticity, and then in mnemonic and cognitive processes (see [2,46]). The most interesting portion of APP is the transmembrane part. The alternative cleavage of APP around this domain can occur by means of α-secretase without the formation of pathological peptides and/or through the concerted action of β- and γ-secretase, resulting in the release into the extracellular space of toxic Aβ peptides with 40 or 42 residues. Aβ42 is more prone to aggregation and is the main component of extracellular insoluble NPs (see [52]). Consistently, the inhibition of Aβ production by using specific inhibitors for β- and γ-secretase or antibodies against Aβ prevents cell death. In addition to causing the formation of extracellular aggregates, Aβ peptides are also present inside neurons (see [52]), in various intracellular sites including the trans-Golgi network [53], the endoplasmic reticulum (ER) and the endosomal, lysosomal and mitochondrial membranes (see [52]). In particular, it seems that the accumulation of Aβ within the mitochondria causes mitochondrial dysfunction through the formation of ROS (see [54]). However, although the excessive production of Aβ peptides is observed early in AD patients and is essential for the disease progression (see [54]), it is not sufficient per se to cause full-blown dementia, so much so that some elderly individuals

with significant Aβ burden do not develop cognitive impairment (see [54]) without Tau being present in the affected areas (see below) (see [52]).

Within the same neurons, more or less at the same time as the abnormal demolition of APP takes place, the Tau protein undergoes attacks by proteolytic enzymes and post-translation modifications, of which phosphorylation is the best known [55]. These processes end with its partial demolition and constitution of protein spirals called NFTs. NFTs originate from an abnormal aggregation of the microtubule-associated protein Tau, following its hyperphosphorylation and cleavage (see below, Section 4.4) [55]. Glycogen synthase kinase (GSK-3) and cell division protein kinase 5 (CDK5) are the kinases primarily responsible for phosphorylation of Tau in human brain, and protein phosphatase 2A (PP2A) is the major Tau phosphatase [56]. Caspase 2/3/6/8 and Calpain I/II are the main proteases responsible for the limited proteolisis of Tau [57,58] (see Figure 2). Hyperphosphorylation and truncation cause Tau molecules to move away from microtubules and attach themselves to each other to form NFT lesions (Figure 2).

According to a new study led by scientists at the UC San Francisco Memory and Aging Center, positron emission tomography (PET) brain imaging of pathological Tau protein tangles reliably predicts the position of future brain atrophy in Alzheimer's patients a year or more in advance [59]. This type of brain imaging could open the era of "precision medicine" for the cure of AD. Unlike amyloid, accumulating extensively in the brain and sometimes even in people without symptoms, autopsies of Alzheimer's patients revealed that Tau is concentrated precisely where brain atrophy is most severe and where degeneration will occur.

As previously reported, extensive studies focusing solely on the independent reduction in the neurotoxicity of Aβ or that of Tau have not shown significant translational efficacy in the treatment of AD (see [2]). Therefore, the reciprocal interaction between Aβ and Tau in mediating cognitive dysfunction in AD patients has to be taken into account. In a recent review, Zhang et al. [2] summarize interesting data on the interplay between Aβ and Tau in promoting neurodegeneration. We limit ourselves to reporting the conclusions as summarized and proposed by the authors: "(A) Aβ drives tau pathology by inducing tau hyperphosphorylation, and pTau mediates toxicity in neurons. (B) Tau mediates Aβ toxicity, and Aβ toxicity is critically dependent on the presence of tau. (C) Aβ and tau target cellular processes or organelles synergistically and may amplify each other's toxic effects. (D) Aβ and tau may coexist in pathological locations". We add to point (A) that also, Tau cleavage mediates toxicity in neurons, as is discussed later in the paper.

### 3.2. Mitochondrial Dysfunction and Oxidative Stress Are Early Features of AD Brains

Mitochondrial dysfunction, as well as the presence of extensive OS, are pivotal and early features of AD [60,61]. In addition, another early and pre-clinical phenomenon which precedes pathological changes in the AD brain by decades is glucose hypometabolism, now considered a pathophysiological feature of AD, rather than a consequence of it; it seems possible to improve the abnormal brain energy metabolism by correcting the reduction in thiamine diphosphate in AD, closely associated with glucose metabolism [62].

However, here, we only analyze the myriad of complex interactions occurring during the AD development between ROS and the two pathological proteins involved, Aβ and Tau. Preferably, our gaze focuses on the Tau protein (see Section 4), to understand the importance of Tau–ROS cooperation in driving the disease progression.

As mentioned above, Aβ and Tau target cellular processes or organelles in a cooperative and/or synergic way, thus amplifying each other's toxic effects. The cellular compartment in which Aβ and Tau 'work' as a pair to damage neuronal viability is the mitochondrion, especially those located at synapses [63,64]. Thus, it is not remarkable that the mitochondrial dysfunction is causally implicated synaptic deterioration which takes place early in the etiology of AD, mainly through the generation of deleterious ROS [65].

Specifically, post mortem analyses of the brains of AD patients revealed a significant reduction in pyruvate dehydrogenase (PDH), ATP citrate lysate and ketoglutarate dehydrogenase (see [66]), although the main alteration of energy metabolism found in AD is the failure of OXPHOS (see [66]), accompanied by a reduction in the efficiency of electron transfer along the mRC, resulting in an increase in ROS production mainly at the level of complex I and complex III. The mitochondria themselves are vulnerable to OS, which generates a further increase in ROS levels, leading to cell death by disrupting the bioenergetic functions of these organelles (see [63,67]). Similarly, reduced activity of complex IV (cytochrome *c* oxidase, COX) was also observed in the occipital, parietal, temporal and frontal lobes, as well as in the hippocampus of patients with AD [68,69].

In particular, in experiments on isolated mitochondria incubated in the presence of unaggregated Aβ, this peptide turned out to be capable of inhibiting COX, likely by establishing a direct bond with the heme-a group to form an Aβ–heme complex which also exhibits peroxidase activity [70]. In other subsequent studies, we found that Aβ causes a selective defect in Complex I activity associated with an increase in ROS and an impairment of COX, likely due to the deleterious action of ROS [71], i.e., due to peroxidation of membrane lipid, i.e., cardiolipin, by which COX activity is strictly dependent [72]. Our investigation on the effect of Aβ on mitochondrial respiratory function was carried out by using homogenate from CGCs, treated with low sub-toxic concentrations of Aβ (see Figure 1). After that, Alzheimer's synaptic-enriched brain samples were also analyzed with the same result, thus confirming that the above-mentioned experimental in vitro approach represents a valuable tool to investigate how Aβ induces the impairment of specific mitochondrial enzymes in neurons [71], in a way actually corresponding to what happens in vivo in the brains.

Interestingly, Aβ peptide is detectable in the membranes of mitochondria [73] of post mortem AD brains, in the AD brains of living patients and in the brains of transgenic AD mice [74] and it is involved early in the ROS generation at the mitochondrial level, prior to amyloid plaque formation [74,75].

The excessive ROS generation results in a change in mitochondrial permeability due to the opening of the transition pore (mPT), leading to neuronal damage into selective, particularly susceptible populations of brain (i.e., hippocampus and neocortex) with the manifestation of classical AD clinical symptoms [76]. One line of evidence indicated that the interaction of Aβ with mitochondrial proteins, including major components of mPT, exacerbates mitochondrial stress in AD mouse models (for refs, see [77]). Increased formation of the Aβ–cyclophilin D (Cyp D) complex—where Cyp D is a matrix component of mPT—has been reported in the brain mitochondria of AD patients and in transgenic mice [77,78]; it activates the apoptotic cascade and ultimately causes synaptic failure (see [79]).

Noteworthy is the redox interplay between mitochondria and peroxisomes occurring in mammalian cells, including primary neurons, on cooperative fatty acid β-oxidation, redox interplay and organelle dynamics. Thus, it is not surprising that a growing body of evidence highlights the pivotal role of peroxisome alterations in triggering oxidative stress and mitochondrial deficits associated with the AD progression [80–83].

As anticipated, the brain is very vulnerable to OS [84,85] (see Section 2.1). Extensive evidence shows that OS also occurs before the onset of symptoms in AD and that oxidative damage is detected not only in susceptible regions of the brain [61] but also in peripheral areas (for refs, see [61]). Therefore, since OS begins very early in the onset of the disease, any treatment is likely to be more effective if started early, before other harmful processes take over in a "point of no return".

Recent studies have shown that the onset of AD is commonly preceded by an intermediate prodromal stage known as Mild Cognitive Impairment (MCI). In MCI, there is no significant increase in either NPs or in NFTs [61], but decreased levels of non-enzymatic AOX—such as uric acid, vitamin C, vitamin E, vitamin A, lutein, zeaxanthin and α-carotene—and reduced activity of AOX enzymes, such as SOD, glutathione peroxidase (GPX) and glu-

tathione reductase (GR) [61,86] are detected. This widespread oxidative damage preceding the classical neuropathological changes which characterize the AD phenotype of MCI suggests that oxidative imbalance not only appears in the early stage of AD pathogenesis, but also it plays a key role in disease onset and progression.

OS markers (advanced glycation end products, ROS, lipid peroxides, hydroxyl radical adducts with DNA bases and carbonyls) increase in the brains of AD patients, in particular into the hippocampus and cortex (for refs, see [61]), and also in experimental animal models of AD [33,87]. Several studies have confirmed high levels of 4-hydroxinonal (4HNE) and malondialdehyde (MDA) in the brains of MCI and AD patients at autopsy [88,89], as well as in blood samples from AD patients (for refs, see [90]). Protein oxidation, classically evidenced by an increase in the levels of carbonylated proteins, is also detectable in the hippocampus and in the parietal cortex of AD brains [91]. Augmented levels of oxidized bases (i.e., 8-oxo-2 dehydroguanine, 8-hydroxyadenine, 5-hydroxyuracil), indexes of nuclear and mitochondrial DNA oxidation, were found in the temporal, parietal and frontal lobes (for refs, see [64,92]). Interestingly, oxidative DNA damage was found in AD subjects not only associated with the most vulnerable regions of the central nervous system (CNS), but also in their peripheral blood cells [93].

Additionally, the accumulation of Aβ and Tau pathoproteins, along with mitochondrial dysfunction, further intensifies OS in the brain. In fact, that there is a clear relationship between Aβ and OS is an established fact: elevated levels of Aβ1-40 and Aβ1-42 are concomitant with increased levels of oxidation products from proteins, lipids and nucleic acids in the hippocampus and AD cortex [94]. In contrast, regions of the brain with low Aβ levels (e.g., the cerebellum) did not have high concentrations of OS markers [64,95]. These effects, which were observed prior to any plaque deposition, were found in the brain of the triple transgenic mouse model of AD, in which lipid peroxidation (LPx) increased while GSH and vitamin E levels contextually decreased [96]. Treatment with Aβ also leads to lower levels of GSH and increased levels of markers for apoptosis, as well as TRX-1 and GR, two antiapoptotic proteins turning into their oxidized forms in response to treatment [97]. Consistently, more HNE bound to the proapoptotic protein p53 was found in patients with inferior parietal lobule AD than in controls [98]. Post mortem investigations also revealed that reduced GSH and inversely increased GSSG levels in both patients with MCI and severe AD are correlated with cognitive function decline [99]. In particular, GSH is considered a biomarker for the early stages of AD since the ratio of GR to GPX activity is diminished in MCI and further decreased in the late stages of the disease [100], suggesting that a progressive reduction in the ability to recycle GSH occurs during the disease progression.

Interestingly, ROS overproduction, consistently with the activation of lipid peroxidation, also causes changes in oxidized forms of cholesterol, i.e., oxysterols, including 7-ketocholesterol and 7β-hydroxycholesterol, which were detected in the frontal and occipital cortex of AD brains during disease progression [101]. Moreover, Burlot et al. observed that normalizing the level of another oxysterol, such as 24S-hydroxycholesterol, significantly improves the synaptic alterations, the reduction in dendritic length and spine density and memory in the hippocampus of THY-Tau22 mice, a model of AD-like Tau pathology without amyloid pathology [102].

Furthermore, in both AD patients and transgenic mice, the interaction of Aβ with alcohol dehydrogenase, known as ABAD (Aβ binding alcohol dehydrogenase), in mitochondria [103] causes increased ROS formation, mitochondrial dysfunction and finally, apoptosis [104,105]. Experimental evidence has suggested that ABAD overexpression increases ROS production in APP transgenic mice [103]; in parallel, the inhibition of Aβ-ABAD attenuates OS [106].

Other experiments have shown that OS can cause both increased production and accumulation of Aβ [100,107], thus also supporting the inverse relationship. In a mouse model expressing a double APP mutant, OS induction aggravated plaque pathology and increased Aβ levels (1–42) [108]. Consistently, the overexpression of SOD-2 induces a reduction in plaque-Aβ deposition without affecting the levels of soluble and fibrillar

Aβ [109]. The introduction with the diet of nutraceuticals with AOX action, such as epigallocatechin-3-gallate, which is present in large quantities in green tea, inhibits the formation of Aβ amyloid aggregates [110]; curcumin reduces the density of NPs by about 40% [111]; vitamin C inhibits Aβ fibrillation [112]. We refer to our recent review [113] for further information.

## 4. Tau–ROS Interplay

"According to a generally accepted metaphor, Aβ peptides represent the trigger of a gun, but the bullet that kills is the toxic derivative of TAU", explains Professor Pietro Calissano, president emeritus of EBRI, the brain research foundation created by Rita Levi Montalcini, of whom Calissano was the closest collaborator for forty years. By taking a cue from the Nobel Prize discoveries, Calissano's team has developed, after a decade of experimental work on transgenic animals that develop Alzheimer's syndrome, a monoclonal antibody (12A12) that selectively blocks a neurotoxic TAU peptide without affecting the healthy long protein and clearly improves the disease manifestations [114,115].

However, let us go step by step.

### 4.1. What Is Tau?

Tau (as already mentioned in the Section 3.1) is a neuron-specific protein that stabilizes microtubules (MTs), mainly in axons. MTs are incessantly assembled and disassembled in cells in a dynamic way thanks to the interaction with Tau. Under physiological conditions, the dynamism of the process is due to the fact that Tau could be phosphorylated or dephosphorylated depending on the cellular balance between the actions of kinases and phosphatases (for refs, see [116]). In some neurodegenerative diseases, including AD, the hyperphosphorylation of Tau modifies its secondary structure and leads to conformational and functional alterations [117], with the first being the loss of the binding capacity to MTs.

The detachment of Tau from the MTs (see [116]) causes the destabilization of the cytoskeletal network and the interruption of axonal transport [116–119]. Furthermore, unbound Tau aggregate to form NFTs which are deposited mainly in brain regions that play an essential role in memory, learning and emotional behaviors—such as the hippocampus, amygdala, entorhinal cortex and basal forebrain—by reducing the number of synapses in these areas (see [90]).

What has been said so far suggests that the study of the Tau protein is—as we see better in the following Sections 4.3–4.5—rather complex.

All the interactions (see [116]) and functions of Tau contribute to the difficulty of understanding how Tau, i.e., the pathologically altered form of protein, mediates neurodegeneration. Among other things, the precise state of the phosphorylation of Tau is difficult to define in the post mortem biopsy material, due to the labile, transient nature of pTau, which rapidly becomes dephosphorylated after excision [120]. Further, since both the onset and progression of AD are dynamic processes that occur over time, often over several years, it is plausible that processes, such as altered Tau aggregation, can produce a series of different effects in the various stages of the disease, which may escape the experimental eye of the most experienced researcher.

### 4.2. Tau Isoforms

Six different isoforms of the Tau protein are expressed in the adult human brain. From a structural point of view, each isoform consists of three domains with distinct roles in promoting the assembly and stability of microtubules within neuronal axons [121]: (i) a projection domain, (ii) a domain rich in proline, which contains abundant phosphorylation sites [122] and contributes to stabilization of microtubules [123], and (iii) a microtubule assembly domain, which consists of repeated conserved motifs of binding to the β-tubulin and comprises the carboxy-terminal portion of the protein. The six isoforms diverge from each other by the number of MT binding domains (3R/4R) [124] and by the presence or absence of one or two projection domains (0N/1N/2N) in the amino-terminal portion

of the protein, that is, the one not directly involved in binding to MTs [125]. Each of the isoforms probably has a particular physiological role as they are differently expressed during development.

### 4.3. Molecular Mechanisms Underlying the Pathology of the Tau Protein

Tau present in pathological lesions undergoes a variety of post-translational modifications, including *O*-glycosylation, ubiquitination, SUMOylation, nitration, glycation, acetylation, isomerization, hyperphosphorylation and proteolytic cleavage or truncation [126–131]. The most relevant alterations associated with the onset and progression of AD are the hyperphosphorylation and truncation of Tau (see [131]). In particular, both phosphorylation and truncation reduce the affinity of Tau for microtubules which undergo disassembly with consequent compromission of axonal transport [45,132]. One of the current hypotheses explaining neurodegeneration in AD suggests that these changes are—in turn—the consequence of several events in which increased Aβ production, OS burst and relentless mitochondrial dysfunction facilitate the hyperphosphorylation and the cleavage of Tau, converting this protein into a toxic species with a negative impact on neuronal vital function [45].

### 4.4. Tau Hyperphosphorylation and Truncation

Normal Tau is a phosphoprotein; thus, its phosphorylation plays an important role in the regulation of its physiological function [133]. Conversely, pathological phosphorylation at specific sites significantly increases its propensity to detach from microtubules and aggregate (for refs, see [134]). The question of why Tau proteins become hyperphosphorylated remains unanswered. Mitochondrial dysfunction and ROS explosion [135,136], as well as catalysis by Aβ42 [137], and also glucose-related metabolic disorders such as diabetes [138], are mentioned as suspicious factors for the initiation of Tau hyperphosphorylation in AD.

Abnormal phosphorylation drastically reduces the affinity of Tau for β-tubulin [139], with consequent microtubule depolymerization and pathological aggregation of Tau (see [140,141]). The progressive accumulation of NFT disrupts synaptic activity and neuronal signaling [121,142], resulting in memory/learning impairment and cognitive decline.

Overall, Tau phosphorylation increases by approximately four times in AD compared to healthy brains [134].

The phosphorylation state of Tau proteins is mediated by the coordinated interaction between a variety of kinases; the main ones involved are GSK-3, CDK5, the activated protein kinase from AMP, protein kinase A, and FYN (for refs, see [141,142]), and phosphatases, including phosphatase-1 (PP1) and PP2A [143]. However, the contribution of these enzymes in neurodegeneration remains to be established [144].

Additionally, truncation is another process which critically contributes to AD pathogenesis due to its ability to promote the propensity of Tau for aggregation, ending up with the formation of NFTs (see [145]).

The Tau protein is a substrate of several endogenous proteases (see [116,145]). In addition to being cleaved at the carboxy-terminal region [146], preferably by caspase-3 to aspartic acid 421 (Asp 421) [116], Tau is also cleaved at the N-terminal domain with the generation of a diagnostic peptide of 20–22 kDa mapping between 26 and 230 amino acid residues of full-length protein (NH$_2$hTau), which preferentially accumulates into mitochondria-rich synapses from the hippocampus and frontal cortex of AD subjects. This NH$_2$hTau fragment is causally associated with neurofibrillar degeneration and synaptic impairment in the human brain affected from AD [14,145,147]. Therefore, N-terminal Tau truncation participates in disease progression and is a critical step in the toxic cascade leading to neuronal death, similar to that proposed for C-terminal Tau cleavage by caspases (see [116]).

In the following paragraphs, we provide an overview of experimental results, highlighting the importance of Tau proteolysis in its N-terminal projection domain and discussing the potential clinical and translational impact of the antibody-mediated neutralization of NH$_2$hTau fragments in different in vitro and in vivo paradigms.

### 4.5. Hyperphosphorylation of the Tau Protein Is Essentially Interlinked with the OS. Aβ Production and Mitochondrial Dysfunction Participate in the Process

Disturbances in the delicate balance between ROS and AOX defenses (see [131]), particularly fragile in brain neurons, are suspected to play a crucial role in modulating Tau phosphorylation and truncation.

In support of this finding is the discovery that Tau-overexpressing cells show increased vulnerability to OS and reduced viability [148,149].

Here, we examine the complex relationship between the accumulation of toxic Tau species and OS along a vicious cycle that leads to a progressive increase in both ROS and abnormal Tau and, finally, cell death in neurodegenerative Tauopathies. At the same time, we collect scientific data concerning the elimination/scavenging of ROS in order to understand whether it mitigates the accumulation of the toxic Tau peptides and then improves mitochondrial and cognitive functions in AD.

The Tau phosphorylation process is dependent on the activity of two enzymes having opposite activity, i.e., kinase and phosphatase (see above). Therefore, it is plausible to hypothesize that OS plays a role in the activation of the protein kinase and in the suppression of phosphatase, in particular the predominant Tau-directed GSK-3β and PP2A, respectively (see Figure 2). Not surprisingly, GSK-3 activity is upregulated under OS [150] (see Figure 2). There are two GSK-3 isoforms, GSK-3α and GSK-3β, which are highly homologous and have been implicated in a variety of critical regulatory roles [151]. It seems that of the two GSK-3 isoforms, GSK-3β is highly expressed at neuronal synapses and is involved in synaptic plasticity. GSK-3β is a constitutively active kinase and its activity is primarily regulated by the phosphorylation status of its Ser-9 residue. Phosphorylation of the Ser-9 residue inhibits GSK-3β activity, whereas dephosphorylation activates GSK-3β (see [152]. Now, for simplicity, we discuss the functions of GSK-3 without discriminating between the two isoenzymes, even though most studies of GSK-3 refer to GSK-3β.

To this regard, it was recently observed that in PC12 neuron cells exposed to 100 μM H$_2$O$_2$, treatment with low doses of GSK3β inhibitors protected them from H$_2$O$_2$-induced OS and apoptosis. Conversely, higher concentrations of GSK3β inhibitors induce opposite effects related to apoptosis and Tau phosphorylation, demonstrating that fine modulation of this kinase activity actually prevents apoptosis, as well as OS-induced Tau phosphorylation [153].

In parallel, increased expression of GSK-3β and the p25 activator of cyclin-dependent kinase 5 pause the mitochondrial movement in cortical neurons along the neuritic process [154]. Similarly, GSK-3β inhibition reverses axonal transport disrupted by Tau overexpression in Drosophila [155]. All results indicate that GSK-3β is crucially involved in the regulation of mitochondrial translocation under the condition of Tau overexpression. Furthermore, the treatment of cultured neuronal cells with ROS mimicking mitochondrial OS promotes Tau phosphorylation [156] by increasing the GSK-3β activity [157].

According to published experiments and reports, therefore, OS—along with mitochondrial dysfunction and involvement of Aβ—is interconnected with Tau pathology. However, equally, as we see, the pathology of Tau induces OS and mitochondrial damage [52], in part by modulating the toxicity of Aβ.

### 4.5.1. OS-Induced Tau Phosphorylation

Experimental data show that the pro-oxidant inhibition of GSH synthesis with butionine sulfoximine determines, in an in vitro model of chronic OS, an increase in Tau phosphorylation in neuronal cultures [131,158], as well as increases in the levels of Tau phosphorylation in the epitope of coupled helical filaments (PHF-1) (serine 396/404) [152]. Similarly, chronic OS elevated the phosphorylation of Tau at specific AD-like phosphoepitopes in cultured neural cells [152]. Further, in primary rat cortical neuronal cultures

stimulated by the combination of the copper chelator, cuprizone and OS ($Fe^{2+}/H_2O_2$), Tau phosphorylation is significantly enhanced by elevated GSK-3 activity [157]. In addition, a reduction in cytoplasmic SOD1 or a deficit in mitochondrial SOD2 [128] increase the profile of Tau phosphorylation in Tg2576 AD transgenic mice expressing human mutated APP. In this same study, by crossing SOD2 ($-/-$) animals into Tg2576 genetic background, the authors generated double transgenic mice lacking SOD2 with marked cerebral accumulation of Aβ peptide and elevated levels of pTau [135,159].

Taken together, these data support the crucial involvement of ROS in determining Tau toxicity. Yet, another possible link between OS and pathological phosphorylation of Tau appears to be mediated by carbonyl-4HNE, a product of LPx, which facilitates the aggregation of pTau in vitro and induces the hyperphosphorylation of Tau (see [131]).

### 4.5.2. Tau Phosphorylation-Induced OS

Otherwise, that the Tau protein can effectively induce the production of deleterious ROS is evidenced by compelling in vitro and in vivo studies. The tight relationship between Tau pathology and OS has been proven in P301L and P301S, two lines of transgenic mice carrying the human Tau gene with the P301L or P301S mutation linked to frontotemporal dementia. The dramatic accumulation of pTau, which causes neurodegeneration and the development of insoluble NFTs [160], has been observed in the brains of these animals. In particular, hippocampal Tau phosphorylation in transgenic Tau mice with the P301L mutation induces mitochondrial dysfunction, resulting in $H_2O_2$ production, LPx and ultimately neuronal loss [90,161,162]. Likewise, the brains of a P301S human Tau mouse model exhibit huge OS into nerve cells and strong upregulation in the levels of carbonyl proteins in cortical mitochondria.

Interestingly, the application of extracellular Tau at different aggregation stages to cortical co-cultures of neurons and astrocytes showed that only insoluble Tau aggregates induce ROS production by activating the calcium-dependent Nicotinamide Adenine Dinucleotide Phosphate (NADPH) oxidase [163].

Consistently with what has been said so far, in recent years, AOX therapy has received considerable attention as a promising approach to slow down the progression of AD, thus reinforcing the hypothesis that oxidative damage may be responsible for cognitive and functional decline in AD patients.

Interestingly, under conditions of OS, neurons have the intrinsic compensatory ability to generate AOX enzymes to counteract damage (see [45]). Inducible expression of the genes encoding these detoxifying enzymes is controlled by Nrf-2 (see [43]), the main regulator of redox homeostasis (see [45,164]). In this regard, the pharmacological use of sulforaphane, a potent activator of the Nrf-2 signal transduction pathway, significantly reduced the level of abnormal Tau both in hippocampal neuronal cultures and in immortalized cortical cells (CN1.4) [165]. In the same context, complementary studies showed that sulforaphane treatment reduced memory loss [166].

Demethoxycurcumin has been proved to inhibit the phosphorylation of Tau pS262 and pS396 in murine neuroblastoma N2A cells [167]. Curcumin reduced pTau levels and increased Heat Shock Proteins, molecular chaperones involved in Tau clearance [168]. Furthermore, there is also an association between beta carotene and Tau in patients with AD [169]. Other experiments have shown that the active component of Ginkgo biloba, ginkgolide A, inhibits GSK3β and suppresses the level of Tau phosphorylation [170]. Other AOX, such as vitamins E and C [171,172], also have a protective effect against Tau-mediated neurotoxicity.

### 4.6. Pathological Tau Affects Mitochondrial Function in AD

Tau hyperphosphorylation negatively impacts mitochondrial bioenergetics and turnover (changes in number, size and distribution), as shown in AD post mortem brains and rodent models and in cell culture studies, as well as transgenic mice: pTau causes a reduction in ATP generation, dissipation of mitochondrial membrane potential (mtΔΨ), promotion of

mitochondrial fission and fragmentation [78,173], as well as an increase in OS [121], essentially due to Tau-induced inhibition of the activity of mitochondrial complex I. Pathological Tau alters mitochondrial dynamics by regulating mitochondrial fission/fusion proteins, moving towards excessive fission [174,175]. In particular, it has been documented that the abnormal interaction between pTau and Dynamin-like protein 1 (DLP-1 or Drp1), i.e., the protein that mediates the membrane fission, causes the degeneration of mitochondria and synapses in the brain tissues of APP, APP/PS1 and 3xTg-AD mouse models. Similar findings found in the brain tissues of AD patients by Manczak and Reddy [176] further support the validity of the role of pTau in impairing mitochondrial dynamics and respiration.

There is also much debate on the possible role of the mitochondrial permeability transition pore (mPTP) in participating to aggravate the mitochondrial damage [177] caused by Tau pathology and its impact on the deterioration of synaptic function in AD pathogenesis (see [45]). In particular, studies carried out on the involvement of mPTP have shown that mitochondrial fragmentation, depolarization and mitochondrial calcium management defects induced by the constitutive expression of truncated Tau can be prevented by using the drug Ciclosporin A [178], a compound that prevents the opening of mPTP by inhibiting the binding of Cyp D to other mPTP elements [45,179]. These findings clearly indicate that truncated Tau along with pTau are more likely to critically contribute to early mitochondrial damage reported in brain samples and in AD neuronal in vitro models [180].

Consistently, in another cell system represented by CGCs exposed to low, sublethal concentrations of toxic Tau fragment, we found that N-terminal-truncated Tau altered the ATP synthesis and the membrane potential [9,14], thus confirming that this peptide is able to induce alterations in mitochondrial bioenergetics. The intracellular bioavailability of ATP appeared to be significantly reduced following treatment with the synthesized $NH_2$-26-44 peptide—which is the minimal active fraction that retains the deleterious effect in vitro of the longer overexpressed parental $NH_2$-26-230 human Tau fragment [7,8,12,145], but not with its biologically harmless counterpart, $NH_2$-1-25 [7,8,12,145]. In particular, the specific effect derives from the block in the thiol group/s located in the active site of ANT-1, a component of mPTP [9,23] (refer to the Section 1) (see Figure 1). These results strongly support the hypothesis that an aberrant activation of caspases, following apoptotic stimuli and/or other neurodegenerative insults, can produce one or more toxic fragments of Tau derived from the N-terminal domain, which further contribute to propagate and increase the cell damage during AD progression (see [146]). Finally, N-terminal-derived Tau fragments, including $NH_2$-26-230 and $NH_2$-26-44 peptides, (i) are present at high levels in synaptic compartments from cell models, experimental AD animals and post mortem human brains of AD subjects [181]; (ii) contribute to synaptic deterioration and unbalanced neurotransmission [9,182]; (iii) are also distinguishable in cultured neurons in the hippocampus and organotypic slices exposed to low concentrations of extracellular Aβ oligomers and in 3xTg mice carrying mutated human APPSwe, TauP301L, PS1M146V, i.e., in vitro and in vivo AD systems (see [146]) characterized by marked mitochondrial and synaptic disability.

### 4.7. Aβ and Tau Act Synergistically on Cellular Processes or Organelles and Can Amplify Each Other's Neurotoxic Effects

What cannot be overlooked in this study report is the synergistic way in which the two toxic forms of proteins, Tau and Aβ, interact on the mitochondria. Here, we cover this issue, albeit briefly.

First of all, the topic that has been on the research bench for years concerns a crucial question, i.e., the toxicity of Aβ 'also' depends on Tau. Several observations have led to this conclusion. First of all, it can be said that Aβ-induced neurotoxicity is mediated not only by Aβ overdeposition but also by protease(s) activation. It would be these proteases that generate neurotoxic fragments of Tau [2]. Therefore, as reported by George S. Bloom, Aβ acts as a trigger and Tau can be the bullet [183]. In support of this finding, the recombinant expression of the human Tau protein in Tau-depleted neurons restores the neurons' sensitivity to Aβ toxicity. Furthermore, increased Tau levels can inhibit APP

transport in axons and dendrites, suggesting a direct causal link between Tau and APP in defects of axonal transport [184,185]. Furthermore, intracerebral injections of Aβ amplify pre-existing Tau pathology in several transgenic mouse models [185,186], while the lack of Tau revokes the toxicity of Aβ [185,187].

The cellular compartment where Aβ and Tau work together in damaging neurons is the mitochondrion [188]. A close relationship has already been established between mitochondrial damage and Tau, on the one hand, and Aβ on the other. However, is it possible that these two molecules synergistically influence mitochondrial integrity and functions? Additionally, how do they relate to each other?

Several studies suggest that Aβ aggregates and pTau can block mitochondrial transport from somatic compartment towards the distal energy-demanding synapse, leading to the starvation of terminal ends, local increased susceptibility to ROS damage, alteration in neurotransmission and then neurodegeneration [189]. Furthermore, in triple transgenic mouse models combining Aβ and Tau pathologies (pR5/APPSw/PS2 N141I)—triple AD mice—the deregulation of complex I activity was found to be strictly dependent on Tau, while the inactivation of complex IV was dependent on Aβ. Besides, bioenergetic deficits occur early in 3xTg-AD triple transgenic mouse model (P301LTau/APPSw/PS1M146L) [190], which show an age-related decrease in enzyme activity regulators of OXPHOS, such as COX, or PDH [191], with increased OS and LPx (see [149]).

Collectively, this compelling evidence consolidates the idea that a synergistic effect of Tau and Aβ contributes to the pathological weakening of mitochondria in an early stage of AD.

### 4.8. 12A12: A Monoclonal Antibody Able of Specifically Binding the Toxic Tau

Following morphological, biochemical and functional studies that led to the identification of the most toxic Tau-derived peptide, a monoclonal antibody, 12A12 (12A12mAb), was produced. 12A12mAb can open up new frontiers both in the early diagnosis and in the therapeutic treatment of AD. 12A12mAb is a cleavage-specific neoepitope antibody that selectively binds the neurotoxic 20–22 kDa form of Tau without cross-reacting with its full-length physiological protein [115]. The selectivity of 12A12mAb towards the toxic truncated Tau specie(s) prospects that its therapeutic administration is more likely to be safe in human beings in the absence of undesirable consequences due to the "loss of function" of the normal full-length protein [115,192].

Amadoro's group demonstrated that when this Tau-directed antibody was injected through the tail vein into mice engineered for the development of an AD-like syndrome, i.e., symptomatic 6-month-old Tg2576 and 3xTg mice, it reached the hippocampus in its biologically active form and successfully neutralized/antagonized its target, leading to significant improvement of learning abilities together with a reduction in the cerebral Aβ accumulation and the activation of the inflammatory process, likely through a positive, feed-forward regulation [115]. Recently, the same authors found that the administration of 12A12mAb also exerts a beneficial action on the biochemical, morphological and metabolic parameters associated with ocular damage in the AD phenotype [193].

In this regard, an important goal was to ascertain the presence of the toxic Tau peptide in the retina of mice affected by AD, suggesting its pathological role in visual impairment linked with the disease manifestation. Of note, since the eye belongs to peripheral nervous tissue being a direct extension of the brain, it is not surprising that the retina may be affected by the same neurodegenerative processes that disturb brain function (see [193]). Indeed, visual deficits are common and significant in AD [194,195]. Interestingly, the accumulation of retinal tau in 3xTg mice occurs early and even precedes corresponding pathological changes in the brain [194], consistently with [195]. Considering that the disease progression in the early stages is asymptomatic, eye biomarkers turn out to be boon for early diagnosis of AD.

In this framework, starting from the facts that (i) $NH_2$h-Tau is preferentially located in the mitochondria-rich synapses [14,147,196–198] and (ii) the eye's retina has a high concentration of mitochondria, thus explaining its great metabolic demand, Amadoro's

group recently verified that the rescue of mitochondrial functional alterations in Tg2576 AD mouse retinas is exclusively mediated by Tau-directed immunization with 12A12mAb [193].

Our latest discovery [199] concerns the therapeutic efficacy of 12A12mAb in mice treated with intracerebroventricular (ICV) injection of streptozotocin (STZ), a compound which recapitulates the occurrence of the sporadic, most common form of AD. In line with previous investigations [200–202], we detected a prominent increase in LPx in brains of ICV-STZ mice—which is generally used as an indicator of ROS overproduction—along with concomitant decline in AOX scavenger system(s) and mitochondrial impairment. Importantly, the 12A12mAb i.v. delivery exerts a protective action by mitigating the increase in both cytosolic and mitochondrial ROS, the LPx of the mitochondrial membrane, the decrease in the level of GSH and NADPH and the increase in NADH level, as well as the OXPHOS and the ATP content (see Figure 1). Taken together, these findings indicate that the neuroprotective effect of Tau immunization involves in part the modulation of OS and mitochondrial energetic deficits caused by Tau fragments, known to crucially contribute to the synaptic decline of AD in humans [63].

## 5. Conclusions

Having finally reached the conclusions, the first thing that comes to mind is that the timing and the causal relationship between pathological Tau and OS in AD is like the chicken and egg story. Following age-dependent injuries, mitochondrial ROS overproduction and Tau pathology influence neuronal and synaptic function in a vicious circle, leading to memory loss and cognitive impairment associated with AD progression.

Considering that: (i) Aβ and other risk factors play a crucial role in neuronal OS; (ii) Aβ-mediated mitochondrial OS causes toxic Tau formation in AD brains; (iii) OS impair mitochondrial function, especially in high energy-demanding, post-mitotic neurons, a better understanding of oxidative stress production and strategies to cope with it might offer some novel targets for therapy in the AD clinical management.

The compelling in vitro and in vivo experimental evidence by our research groups [9,14,115], along with the close connection between the toxic Tau form and ROS production in driving neurodegeneration, further strengthen the causal role of N-terminal Tau cleavage in the pathogenesis of human tauopathies, including AD. The specific neutralization of the toxic Tau form by the non-invasive administration of 12A12mAb could be a therapeutic option for affected patients. Furthermore, 12A12mAb might serve as a novel promising tool for an early and non-invasive diagnosis of the disease starting from eye inspection.

**Author Contributions:** Conceptualization, writing—original draft preparation A.A.; visualization A.A.; writing—review and editing G.A.; review and editing D.V. and V.L. All authors have read and agreed to the published version of the manuscript.

**Funding:** This research received no external funding.

**Acknowledgments:** V.L. was supported by Post-Doctoral Fellowship by Operatori Sanitari Associati (OSA).

**Conflicts of Interest:** The authors declare no conflict of interest.

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
