# Peer review of "Role of Oxygen Radicals in Alzheimer’s Disease: Focus on Tau Protein"

_oxygen, doi:10.3390/oxygen1020010_

Round 1

Reviewer 1 Report

This is a good review on the effects of ROS in AD and on its consequence on Tau protein.

The review is well organized and will be usefull to many people.

Few information are missing and must be added with the adequat references. These modifications are mandatory.

The part taken by the mitochondria and the ER in ROS overproduction is well described and illustrated but nothing is said on the peroxisome which is tightly connected to the mitochondria. There are now some arguments supporting that this organelle could contribute to AD. This point must be introduced in the paragraph 3.2. I suggest to add the following references. It is very important: to my opinion the role of peroxisome is underestimated.

  • Lizard G, Rouaud O, Demarquoy J, Cherkaoui-Malki M, Iuliano L. Potential roles of peroxisomes in Alzheimer's disease and in dementia of the Alzheimer's type. J Alzheimers Dis. 2012;29(2):241-54. doi: 10.3233/JAD-2011-111163
  • Zarrouk A, Nury T, El Hajj HI, Gondcaille C, Andreoletti P, Moreau T, Cherkaoui-Malki M, Berger J, Hammami M, Lizard G, Vejux A. Potential Involvement of Peroxisome in Multiple Sclerosis and Alzheimer's Disease : Peroxisome and Neurodegeneration. Adv Exp Med Biol. 2020;1299:91-104. doi: 10.1007/978-3-030-60204-8_8
  • Kou J, Kovacs GG, Höftberger R, Kulik W, Brodde A, Forss-Petter S, Hönigschnabl S, Gleiss A, Brügger B, Wanders R, Just W, Budka H, Jungwirth S, Fischer P, Berger J. Peroxisomal alterations in Alzheimer's disease. Acta Neuropathol. 2011 Sep;122(3):271-83. doi: 10.1007/s00401-011-0836-9
  • Dorninger F, Moser AB, Kou J, Wiesinger C, Forss-Petter S, Gleiss A, Hinterberger M, Jungwirth S, Fischer P, Berger J. Alterations in the Plasma Levels of Specific Choline Phospholipids in Alzheimer's Disease Mimic Accelerated Aging. J Alzheimers Dis. 2018;62(2):841-854. doi: 10.3233/JAD-171036
  • In addition, ROS overproduction, can favor lipid peroxidation which can in turn induce brain damage, and favor beta-amyloid aggregation and deposit and Tau-hyperphosphorylation. These molecules are 7-ketocholesterol and 7beta-hydroxycholesterol. In this context, the following paper must be cited and could be also introduced in the paragraph 2.
  • Testa G, Staurenghi E, Zerbinati C, Gargiulo S, Iuliano L, Giaccone G, Fantò F, Poli G, Leonarduzzi G, Gamba P. Changes in brain oxysterols at different stages of Alzheimer's disease: Their involvement in neuroinflammation. Redox Biol. 2016 Dec;10:24-33. doi: 10.1016/j.redox.2016.09.001. Epub 2016 Sep 16. PMID: 27687218; PMCID: PMC5040635.
  • It is also very important to speak about 24S-hydroxycholesterol which play major roles in AD and which is involved in Tau phosphorylation. The following paper must be also cited
  • Burlot MA, Braudeau J, Michaelsen-Preusse K, Potier B, Ayciriex S, Varin J, Gautier B, Djelti F, Audrain M, Dauphinot L, Fernandez-Gomez FJ, Caillierez R, Laprévote O, Bièche I, Auzeil N, Potier MC, Dutar P, Korte M, Buée L, Blum D, Cartier N. Cholesterol 24-hydroxylase defect is implicated in memory impairments associated with Alzheimer-like Tau pathology. Hum Mol Genet. 2015 Nov 1;24(21):5965-76. doi: 10.1093/hmg/ddv268. Epub 2015 Sep 10. PMID: 26358780.

Author Response

Reviewer 1

This is a good review on the effects of ROS in AD and on its consequence on Tau protein.

The review is well organized and will be useful to many people.

Few information are missing and must be added with the adequate references. These modifications are mandatory.

  • The part taken by the mitochondria and the ER in ROS overproduction is well described and illustrated but nothing is said on the peroxisome which is tightly connected to the mitochondria. There are now some arguments supporting that this organelle could contribute to AD. This point must be introduced in the paragraph 3.2. I suggest to add the following references. It is very important: to my opinion the role of peroxisome is underestimated.

Lizard G, Rouaud O, Demarquoy J, Cherkaoui-Malki M, Iuliano L. Potential roles of peroxisomes in Alzheimer's disease and in dementia of the Alzheimer's type. J Alzheimers Dis. 2012;29(2):241-54. doi: 10.3233/JAD-2011-111163

Zarrouk A, Nury T, El Hajj HI, Gondcaille C, Andreoletti P, Moreau T, Cherkaoui-Malki M, Berger J, Hammami M, Lizard G, Vejux A. Potential Involvement of Peroxisome in Multiple Sclerosis and Alzheimer's Disease : Peroxisome and Neurodegeneration. Adv Exp Med Biol. 2020;1299:91-104. doi: 10.1007/978-3-030-60204-8_8

Kou J, Kovacs GG, Höftberger R, Kulik W, Brodde A, Forss-Petter S, Hönigschnabl S, Gleiss A, Brügger B, Wanders R, Just W, Budka H, Jungwirth S, Fischer P, Berger J. Peroxisomal alterations in Alzheimer's disease. Acta Neuropathol. 2011 Sep;122(3):271-83. doi: 10.1007/s00401-011-0836-9

Dorninger F, Moser AB, Kou J, Wiesinger C, Forss-Petter S, Gleiss A, Hinterberger M, Jungwirth S, Fischer P, Berger J. Alterations in the Plasma Levels of Specific Choline Phospholipids in Alzheimer's Disease Mimic Accelerated Aging. J Alzheimers Dis. 2018;62(2):841-854. doi: 10.3233/JAD-171036

First of all, we thank the reviewer for her/his positive assessment of our review, as regards the observation on the role of peroxisomes in AD neurodegeneration, we have now included a brief discussion on this important issue - which we had left out not out of negligence, but only to focus attention on TAU-ROS-Mitochondria - in  #3.2 paragraph. We have also cited the suggested references.

  • In addition, ROS overproduction, can favor lipid peroxidation which can in turn induce brain damage, and favor beta-amyloid aggregation and deposit and Tau-hyperphosphorylation. These molecules are 7-ketocholesterol and 7beta-hydroxycholesterol. In this context, the following paper must be cited and could be also introduced in the paragraph 2.

Testa G, Staurenghi E, Zerbinati C, Gargiulo S, Iuliano L, Giaccone G, Fantò F, Poli G, Leonarduzzi G, Gamba P. Changes in brain oxysterols at different stages of Alzheimer's disease: Their involvement in neuroinflammation. Redox Biol. 2016 Dec;10:24-33. doi: 10.1016/j.redox.2016.09.001. Epub 2016 Sep 16. PMID: 27687218; PMCID: PMC5040635.

In full agreement with the Reviewer's observation, a mention on the toxic action of two oxysterols, i.e. 7-ketocholesterol and 7beta-hydroxycholesterol, has been added in #3.2 paragraph.

  • It is also very important to speak about 24S-hydroxycholesterol which play major roles in AD and which is involved in Tau phosphorylation. The following paper must be also cited

Burlot MA, Braudeau J, Michaelsen-Preusse K, Potier B, Ayciriex S, Varin J, Gautier B, Djelti F, Audrain M, Dauphinot L, Fernandez-Gomez FJ, Caillierez R, Laprévote O, Bièche I, Auzeil N, Potier MC, Dutar P, Korte M, Buée L, Blum D, Cartier N. Cholesterol 24-hydroxylase defect is implicated in memory impairments associated with Alzheimer-like Tau pathology. Hum Mol Genet. 2015 Nov 1;24(21):5965-76. doi: 10.1093/hmg/ddv268. Epub 2015 Sep 10. PMID: 26358780.

The Reviewer is right! Therefore, also the action of 24S-hydroxycholesterol, a molecule involved in Tau phosphorylation,  has been cited in #3.2 paragraph.

Reviewer 2 Report

The work by Atlante et al. is original  and  written in an appropriate way, and scientifically sound,however it suffers from  self-citing. The amyloid hypothesis dates back to 1984, as the occurrence of the pathology in individuals carrying autosomal-dominant mutations in the genes encoding the amyloid precursor protein (APP) or the γ-secretase complex proteins presenilin 1/2 (PSEN1/2), was observed.  however the idea that  Aβ is the exclusive pathogenic AD factor, has been revised recently   10.1186/s12929-019-0609-7, as number of interventions (vaccines, antibodies, and γ/β-secretases inhibitors) meant to reduce Aβ in the brain, failed to ameliorate cognition [Nat Neurosci 2015, 18: 794–799.], also in pre-clinical settings. many clinical trials on amyloid clearing therapy did not stop or reverse disease progression.

These data as well as the  almost complete failure of clinical trials that experimented anti-amyloid therapies must be cited .

 It is becoming apparent that brain glucose hypometabolism is a pathophysiological feature of AD, rather than its consequence and precedes symptoms by decades (see for example https://doi.org/10.1007/s12264-021-00679-8) https://doi.org/10.1515/revneuro-2017-0006, and Authors appear to ignore the data on the involvement of myelin sheath in the bioenergetics of nervous system.( doi: 10.1098/rsob.200224.) Thjese novel hypotheses should bve discussed in order to give tyhe reade the whole piture of the up-to-date picture about AD.

Author Response

Reviewer 2

The work by Atlante et al. is original  and  written in an appropriate way, and scientifically sound,however it suffers from  self-citing. The amyloid hypothesis dates back to 1984, as the occurrence of the pathology in individuals carrying autosomal-dominant mutations in the genes encoding the amyloid precursor protein (APP) or the γ-secretase complex proteins presenilin 1/2 (PSEN1/2), was observed.  however the idea that  Aβ is the exclusive pathogenic AD factor, has been revised recently   10.1186/s12929-019-0609-7, as number of interventions (vaccines, antibodies, and γ/β-secretases inhibitors) meant to reduce Aβ in the brain, failed to ameliorate cognition [Nat Neurosci 2015, 18: 794–799.], also in pre-clinical settings. many clinical trials on amyloid clearing therapy did not stop or reverse disease progression.

  • These data as well as the almost complete failure of clinical trials that experimented anti-amyloid therapies must be cited.

We thank the Reviewer for his / her appreciation.

The Reviewer is right: we focused the study on Tau-ROS-mitochondria interplay, leaving no space for the therapy used for the treatment of AD.

Now, as suggested by the Reviewer, we have mentioned the near complete failure of clinical trials that experimented anti-amyloid therapies in the revised version.

  • It is becoming apparent that brain glucose hypometabolism is a pathophysiological feature of AD, rather than its consequence and precedes symptoms by decades (see for example https://doi.org/10.1007/s12264-021-00679-8) https://doi.org/10.1515/revneuro-2017-0006, and Authors appear to ignore the data on the involvement of myelin sheath in the bioenergetics of nervous system ( doi: 10.1098/rsob.200224.) These novel hypotheses should be discussed in order to give the reader the whole picture of the up-to-date picture about AD.

Both of the Reviewer's suggestions were included in the revised version.

Reviewer 3 Report

Review of a manuscript “Role of Oxygen radicals in Alzheimer’s disease: Focus on Tau protein” by Anna Atlante and coauthors submitted to “Oxygen”

Protein posttranslational modifications play an important role in the pathogenesis of neurodegenerative diseases. Oxygen radicals are active agents causing protein modifications which are associated with death of neurons and other cells in the brain. The authors of the review submitted to “Oxygen” combined recent findings focused on the interplay between oxygen radicals and formation of pathological forms of Tau protein. They consider a role of harmful species cooperation in the progress of Alzheimer’s disease. This is an important field of biomedical research and the results and hypothesis discussed in the manuscript will be interesting for the readers of the “Oxygen”.

The following corrections should be made:

 Lines 61-62 : ”Some years after, we demonstrated that, in addition to NH2htau fragment, also Abeta 1–42 peptide inhibits the ANT–1-dependent ADP/ATP exchange in a competitive and non- competitive manner, respectively [14].”

This is an awkward sentence which should be corrected as follows “Several years after, we have demonstrated that, in addition to NH2htau fragment, Abeta 1–42 peptide also inhibits the ANT–1-dependent ADP/ATP exchange in a competitive and non- competitive manner, respectively [14]” and also the authors should explain clearly what they mean by “respectively”

Figure 1 Some texts on Figure 1 are small and difficult to read, for example, Alzheimer’s disease brain

Line 103 The authors use here the full name of “oxidative stress”, which is abbreviated as OS on line 40.

Figure 2 In the upper left part of figure 2 a glass of red wine is depicted which contains resveratrol and other polyphenols possessing antioxidant properties and protective effect. Should be corrected.

Lines 138-140 “O2 is the primary ROS produced by metabolic processes. It interacts directly with other molecules through enzymatic or metal-catalyzed processes to produce secondary ROS [for refs see 28].”

The authors should add here the following sentence and the citation: ”Oxidation of proteins by ROS changes their properties and may increase their propensity to aggregate. [Reference Surgucheva et al. γ-Synuclein: seeding of α-synuclein aggregation and transmission between cells. Biochemistry. 2012; 51(23):4743-54. doi: 10.1021/bi300478w.]. This property plays an important role in AD and other neurodegenerative diseases.

Lines 158-159 “Vitamins that show important AOX effects are vitamins C (ascorbic acid) and E.” This is an awkward sentence which should be rewritten as follows:”Vitamins C (ascorbic acid) and E possess high AOX potential”.

Lines 159-160 “Ascorbic acid must be ingested from food (or supplements), especially tomatoes, pineapples, watermelons and all citrus fruits [29].” The sentence should be rewritten as follows: “Ascorbic acid must be ingested from food (or supplements); tomatoes, pineapples, watermelons and all citrus fruits contain the highest amount of vitamin C [29].”

Lines 163-164 ”Just to quickly name a few, even carotenoids, a large class of tetraterpenes, widely distributed among plants, include ROS scavenging among their biological activities [32,33].” Style should be corrected in this sentence.

The same concerns the following sentence: ”Exercise is important, just as a diet that is too high in cholesterol is not good for you.”  The simplest way is just to delete the last five words.

Line 230: ”…Presenilin 2 (PS2), Apolipoprotein E (ApoE).” This should be corrected as follows “…Presenilin 2 (PS2), and polymorphisms in Apolipoprotein E (ApoE).”

Conclusion. Lines 759-763:

“Compelling in vitro and in vivo experimental evidence by our research groups [9,14,101] along with the close connection between toxic Tau form and ROS production in driving neurodegeneration further strengthen the causal role of N-terminal tau cleavage in the pathogenesis of human tauopathies, including AD and indicate that its specific neutralization by non-invasive administration of 12A12mAb could be a therapeutic option for affected patients.”

The sentence is too long and hard to read. Should be split into parts and made easy to understand.

Author Response

Reviewer 3

Review of a manuscript “Role of Oxygen radicals in Alzheimer’s disease: Focus on Tau protein” by Anna Atlante and coauthors submitted to “Oxygen”

Protein posttranslational modifications play an important role in the pathogenesis of neurodegenerative diseases. Oxygen radicals are active agents causing protein modifications which are associated with death of neurons and other cells in the brain. The authors of the review submitted to “Oxygen” combined recent findings focused on the interplay between oxygen radicals and formation of pathological forms of Tau protein. They consider a role of harmful species cooperation in the progress of Alzheimer’s disease. This is an important field of biomedical research and the results and hypothesis discussed in the manuscript will be interesting for the readers of the “Oxygen”.

The following corrections should be made:

  • Lines 61-62 : ”Some years after, we demonstrated that, in addition to NH2htau fragment, also Abeta 1–42 peptide inhibits the ANT–1-dependent ADP/ATP exchange in a competitive and non- competitive manner, respectively [14].”

This is an awkward sentence which should be corrected as follows “Several years after, we have demonstrated that, in addition to NH2htau fragment, Abeta 1–42 peptide also inhibits the ANT–1-dependent ADP/ATP exchange in a competitive and non- competitive manner, respectively [14]” and also the authors should explain clearly what they mean by “respectively”.

First of all, we thank the reviewer for her/his positive assessment of our review.

We agree with the reviewer for his/her observation and in agreement we have changed the sentence.

  • Figure 1 Some texts on Figure 1 are small and difficult to read, for example, Alzheimer’s disease brain

Figure 1 has been changed according reviewer’s suggestion: the font of some texts have been enlarged.

  • Line 103 The authors use here the full name of “oxidative stress”, which is abbreviated as OS on line 40.

The abbreviation, i.e. OS, has been inserted.

  • Figure 2 In the upper left part of figure 2 a glass of red wine is depicted which contains resveratrol and other polyphenols possessing antioxidant properties and protective effect. Should be corrected.

The authors have modified Figure 2, by eliminating the glass of wine - Reviewer is right !! - and replacing it with bottles of hard alcohol.

  • Lines 138-140 “O2 is the primary ROS produced by metabolic processes. It interacts directly with other molecules through enzymatic or metal-catalyzed processes to produce secondary ROS [for refs see 28].”

The authors should add here the following sentence and the citation: ”Oxidation of proteins by ROS changes their properties and may increase their propensity to aggregate. [Reference Surgucheva et al. γ-Synuclein: seeding of α-synuclein aggregation and transmission between cells. Biochemistry. 2012; 51(23):4743-54. doi: 10.1021/bi300478w.]. This property plays an important role in AD and other neurodegenerative diseases.

As suggested by the Reviewer, the sentence and reference have been added.

  • Lines 158-159 “Vitamins that show important AOX effects are vitamins C (ascorbic acid) and E.” This is an awkward sentence which should be rewritten as follows:”Vitamins C (ascorbic acid) and E possess high AOX potential”.

The change has been made.

  • Lines 159-160 “Ascorbic acid must be ingested from food (or supplements), especially tomatoes, pineapples, watermelons and all citrus fruits [29].” The sentence should be rewritten as follows: “Ascorbic acid must be ingested from food (or supplements); tomatoes, pineapples, watermelons and all citrus fruits contain the highest amount of vitamin C [29].”

The change has been made.

  • Lines 163-164 ”Just to quickly name a few, even carotenoids, a large class of tetraterpenes, widely distributed among plants, include ROS scavenging among their biological activities [32,33].” Style should be corrected in this sentence.

The sentence has been consistently rearranged.

  • The same concerns the following sentence: ”Exercise is important, just as a diet that is too high in cholesterol is not good for you.”  The simplest way is just to delete the last five words.

The change has been made.

  • Line 230: ”…Presenilin 2 (PS2), Apolipoprotein E (ApoE).” This should be corrected as follows “…Presenilin 2 (PS2), and polymorphisms in Apolipoprotein E (ApoE).”

The change has been made.

  • Conclusion. Lines 759-763:

“Compelling in vitro and in vivo experimental evidence by our research groups [9,14,101] along with the close connection between toxic Tau form and ROS production in driving neurodegeneration further strengthen the causal role of N-terminal tau cleavage in the pathogenesis of human tauopathies, including AD and indicate that its specific neutralization by non-invasive administration of 12A12mAb could be a therapeutic option for affected patients.”

The sentence is too long and hard to read. Should be split into parts and made easy to understand.

The sentence has been modified.
